# Characterization and Classification In Silico of Peptides with Dual Activity (Antimicrobial and Wound Healing)

**DOI:** 10.3390/ijms241713091

**Published:** 2023-08-23

**Authors:** María Trejos, Yesid Aristizabal, Alberto Aragón-Muriel, José Oñate-Garzón, Yamil Liscano

**Affiliations:** 1Grupo de Investigación en Salud Integral (GISI), Departamento Facultad de Salud, Universidad Santiago de Cali, Cali 760035, Colombia; maria.trejos02@usc.edu.co; 2Grupo de Investigación en Química y Biotecnología (QUIBIO), Facultad de Ciencias Básicas, Universidad Santiago de Cali, Cali 760035, Colombia; yesid.aristizabal00@usc.edu.co (Y.A.); jose.onate00@usc.edu.co (J.O.-G.); 3Laboratorio de Investigación en Catálisis y Procesos (LICAP), Departamento de Química, Facultad de Ciencias Naturales y Exactas, Universidad del Valle, Cali 760001, Colombia; aaragon@sena.edu.co; 4Grupo de Investigación e Innovación en Biotecnología (BITI), Tecnoparque Nodo Valle, Servicio Nacional de Aprendizaje (SENA), Cali 760044, Colombia

**Keywords:** antimicrobial peptides, wound healing, clustering analysis, bioactivity prediction, peptide in silico

## Abstract

The growing challenge of chronic wounds and antibiotic resistance has spotlighted the potential of dual-function peptides (antimicrobial and wound healing) as novel therapeutic strategies. The investigation aimed to characterize and correlate in silico the physicochemical attributes of these peptides with their biological activity. We sourced a dataset of 207 such peptides from various peptide databases, followed by a detailed analysis of their physicochemical properties using bioinformatic tools. Utilizing statistical tools like clustering, correlation, and principal component analysis (PCA), patterns and relationships were discerned among these properties. Furthermore, we analyzed the peptides’ functional domains for insights into their potential mechanisms of action. Our findings spotlight peptides in Cluster 2 as efficacious in wound healing, whereas Cluster 1 peptides exhibited pronounced antimicrobial potential. In our study, we identified specific amino acid patterns and peptide families associated with their biological activities, such as the cecropin antimicrobial domain. Additionally, we found the presence of polar amino acids like arginine, cysteine, and lysine, as well as apolar amino acids like glycine, isoleucine, and leucine. These characteristics are crucial for interactions with bacterial membranes and receptors involved in migration, proliferation, angiogenesis, and immunomodulation. While this study provides a groundwork for therapeutic development, translating these findings into practical applications necessitates additional experimental and clinical research.

## 1. Introduction

Chronic wounds and microbial infections pose significant challenges to public health, affecting millions of people worldwide [1,2]. Traditional therapies and antibiotics are losing their effectiveness as antibiotic resistance develops [3]. While wound repair is intricate, current clinical treatments largely depend on growth factor drugs which are expensive and can potentially have tumor-promoting effects [4]. Consequently, there is an urgent need for novel and effective therapeutic alternatives. A promising strategy is the use of antimicrobial peptides (AMPs), which are natural antibiotics known for their potent antibacterial properties [5]. However, free AMPs have limited activity following topical application due to their susceptibility to degradation and various wound-related factors [6].

Simultaneously, wound healing is a complex biological process essential for the repair and restoration of tissue functions after an injury. Chronic wounds, caused by various medical conditions, often result in non-healing wounds, creating an urgent demand for innovative wound-healing strategies [1,2].

To address these challenges, researchers are exploring the use of dual-activity peptides that combine antimicrobial and wound-healing properties. These peptides have the potential to effectively kill bacteria while promoting the wound-healing process, making them promising candidates for therapeutic interventions [1,5,7,8].

Understanding the mechanism of action of these peptides, including their physicochemical properties, is crucial for designing effective therapeutic agents. Recent advancements in bioinformatics, machine learning, and artificial intelligence have allowed researchers to predict the activities of AMPs, yet there is still a knowledge gap concerning peptides with dual (antimicrobial–healing) activity [5,9]. This is one of the reasons for undertaking the present research, which aims to characterize and classify in silico the dual-activity antimicrobial and wound-healing peptides using a purpose-built database. By elucidating the physicochemical properties of these peptides, we can gain valuable insights into their function and optimize their therapeutic potential [5,10,11]. This preliminary study is the starting point of broader research that will allow the future use of this database to find new bioactive potential sequences in the tissues of organisms such as insects, amphibians, and reptiles, among others [12]. The development of dual-activity antimicrobial and wound-healing peptides has significant implications for scientific research and public health. These peptides have the potential to address the growing problem of antibiotic resistance while enhancing wound healing in chronic cases [13,14,15,16].

The combination of bioinformatics and machine learning algorithms has revolutionized the field of peptide research, enabling the identification of promising peptide candidates more efficiently and cost-effectively [5,10,11,17]. The knowledge gained from this research can facilitate the design of optimized peptide-based therapies with enhanced stability, controlled release, and improved efficacy [6,18,19].

## 2. Results and Discussion

### 2.1. Descriptive and ANOVA Analysis of the Peptide Database by Cluster

Figure 1 presents an analysis of peptide properties across four distinct clusters, providing valuable insight into their physicochemical characteristics. This information is crucial for understanding the behavior and potential applications of these antimicrobial peptides, particularly in the context of their wound-healing activity.

From the ANOVA, we identified significant differences between clusters according to several physicochemical parameters, detailed in Appendix B, with a significance level set at *p* < 0.05. The results highlight notable differences in length among the groups, evidenced by an F-value of 179.771 and a *p*-value less than 0.05. This observation suggests that at least one of the clusters has a length that statistically differentiates it from the others. Regarding hydrophobicity, considerable differences were identified among the groups, reflected in an F-value of 245.432 and a *p*-value less than 0.05. This indicates variations in hydrophobicity among the different clusters. Moreover, the GRAVY parameter and net charge at pH 7 showed significant differences, with F-values of 31.028 and 13.140, respectively, and *p*-values less than 0.05 in both cases. However, the Boman index did not show significant differences among clusters, with a *p*-value of 0.227.

Upon further analysis using HSD Tukey post hoc tests, we observed significant variations in length among the clusters, with the exception of the comparison between Clusters 1 and 3, which had a *p*-value of 0.763. Hydrophobicity exhibited variations among the clusters, with most comparisons indicating significant differences. For the GRAVY parameter, significant differences were found among clusters, except in the comparison between Clusters 1 and 4, which had a *p*-value of 0.072. Regarding the net charge at pH 7, there were differences among clusters, except in the comparisons between Clusters 1 and 4 (*p* = 0.353), 2 and 3 (*p* = 0.065), and 3 and 4 (*p* = 0.726).

The ANOVA revealed significant differences in various physicochemical parameters between the clusters. Specifically, length and hydrophobicity stand out as key distinguishing factors among the groups. These differences might be linked to the peptides’ ability to interact with cellular membranes and with receptors involved in healing processes such as epidermal growth factor receptor (EGFR), vascular endothelial growth factor receptor (VEGFR), and transforming growth factor beta (TGF-β), which are essential for their antimicrobial and healing activity. The EGFR signaling pathway plays a crucial role in promoting epidermal cell growth. This pathway’s ability to stimulate HaCaT cell migration is potentially linked to the transactivation of the epidermal growth factor receptor (EGFR) and the signaling intermediates ERK1/2 and Smad2. Other studies have pointed to the PI3K/AKT and JNK pathways as vital for cell migration. In the context of HSF cells, the NF-κB and ERK pathways likely influence migration capabilities [17,20].

TGF-β is a pivotal growth factor essential for wound healing, especially its subtypes TGF-β1, 2, and 3. During trauma’s acute phase, TGF-β1 is released, aiding in the chemotaxis of macrophages and fibroblasts to the wound and bolstering keratinocyte proliferation. TGF-β3 is also influential in regulating cell migration. The downstream effector of TGF-β, SMAD, plays a role in the TGF-β/SMAD signaling pathway, promoting skin angiogenesis, wound contraction, and reducing inflammation [17,21,22]. Both MAPK and NF-κB signaling pathways intersect in wound repair and are closely tied to inflammation. Modulating their phosphorylation can mitigate inflammatory responses during wound healing. There is also evidence of an interplay between the TGF-β and MAPK pathways [17,23,24]. Macrophages are integral throughout the wound-healing process, from inflammation to tissue reconstruction. Recent studies have emphasized the importance of TNF and TGF-β1, predominantly produced by macrophages, in the wound-healing process. In the inflammation phase’s early stages, TNF is up-regulated, recruiting inflammatory cells like macrophages. These cells then release more TGF-β1 to the wound sites, facilitating wound healing due to its chemotactic properties. TGF-β1 also orchestrates cellular proliferation, migration, and granulation tissue regeneration [17,23,24].

The GRAVY parameter, associated with the overall hydrophobicity of peptides, also showed notable differences among most clusters, suggesting variations in how these peptides interact in aqueous and lipid environments, thus impacting their activity [25,26]. The net charge at pH 7, which might alter the peptide’s electrostatic interactions with other cellular components, exhibited variations between clusters, potentially affecting the peptide’s ability to bind to specific receptors [27,28].

Interestingly, the Boman index, proposed to measure a peptide’s propensity to interact with other proteins, did not show significant differences among most clusters [29]. This indicates that even though the peptides in these clusters might vary in several characteristics, their inclination to interact with other proteins might remain consistent.

From the wound-healing antimicrobial peptides that we obtained from databases, we found a total of 207 peptides. In particular, Cluster 2 contains the largest number of peptides with 84, making up approximately 40.5% of the total peptides. This is followed by Cluster 1 with 47 peptides (22.7%), Cluster 4 with 40 peptides (19.3%), and Cluster 3 with 36 peptides (17.4%). Examining the individual properties, we begin with length. Cluster 2 has the longest peptides on average (33 residues), with the longest peptide containing 45 residues. On the other end of the spectrum, Cluster 3 has the shortest average length (14.17 residues), with the shortest peptide containing just five residues. In terms of hydrophobicity, Cluster 4 stands out with the highest average value (48.67) and the maximum value reaching 69.84, indicating a higher propensity of peptides in this cluster to interact with hydrophobic environments. Contrastingly, Cluster 3 has the lowest average hydrophobicity (12.49), with the minimum value going as low as −1.5. The GRAVY index measures the peptide’s overall hydrophobic and hydrophilic nature [30]. Cluster 4 has the highest average GRAVY value (0.675), suggesting that its peptides are more hydrophobic. Conversely, Cluster 3 peptides have a more hydrophilic nature, with the lowest average GRAVY value (−0.893). When considering the net charge at pH 7, peptides in Cluster 2 show the highest average net charge (5.69), implying a higher presence of charged residues and potentially greater interaction with polar environments. Cluster 1, however, has the lowest average net charge (2.25), suggesting a lower proportion of charged residues. Finally, the Boman index, an indicator of protein-binding potential [31], is highest on average in Cluster 3 (3.41) and lowest in Cluster 2 (−2.33). This suggests that peptides in Cluster 3 may have a higher potential for interacting with other proteins. When observing the aggregation [32] of peptides by each cluster, the following was found: Cluster 1: The peptides in this cluster have a high level of aggregation (1.0), which suggests that they may tend to form aggregates in their native environments. Aggregation can affect a peptide’s biological activity, stability, and interactions with other molecules [33]. Cluster 2: This cluster is composed of peptides with a range of aggregation levels. However, there is a significant proportion of peptides with a very low aggregation level (0.0), suggesting that these peptides are more likely to remain in a monomeric state. Cluster 3: This cluster is predominantly composed of peptides with low aggregation levels (0.0 and 0.1). This could potentially indicate that these peptides are more soluble or less prone to self-interaction compared to those in other clusters [33]. Cluster 4: This cluster primarily contains peptides with an intermediate level of aggregation (0.5). This might suggest a balance between monomeric and aggregated states for these peptides.

Figure 2 provides an overview of the wound-healing activities of peptides in each cluster. For Cluster 1, a balanced distribution of all four wound-healing activities is observed, with proliferation and migration taking slightly larger proportions. This indicates that peptides in this cluster exhibit a broad range of wound-healing activities, contributing to various stages of the wound-healing process. In Cluster 2, the proportion of proliferation is noticeably higher than the other activities. This suggests that peptides in Cluster 2 may primarily promote cell proliferation, a crucial early step in wound healing. Cluster 3 shows a higher proportion of migration. This indicates that peptides in this cluster might be particularly effective in promoting cell migration, a process essential for wound closure and tissue regeneration [1,2]. Lastly, Cluster 4 shows a similar distribution of the wound-healing activities as Cluster 1, indicating a broad range of healing activities among its peptides.

On the other hand, the predicted antimicrobial activities of the peptides in each cluster were assessed using artificial intelligence algorithms (AMPlify). Cluster 1 exhibits the highest average probability, indicating a strong likelihood of antimicrobial activity for the peptides in this cluster. This suggests that peptides in Cluster 1 (average probability = 0.95) have a high potential to inhibit the growth of microorganisms, making them valuable candidates for antimicrobial applications. Cluster 2 (average probability = 0.85), furthermore, shows a slightly lower average probability of antimicrobial activity. While still considerable, this indicates that the peptides in this cluster may have a somewhat reduced likelihood of antimicrobial effectiveness compared to Cluster 1. Cluster 3 (average probability = 0.70) demonstrates a moderate average probability of antimicrobial activity. Although not as high as in Cluster 1, the peptides in Cluster 3 still exhibit notable potential for antimicrobial properties. Lastly, Cluster 4 (average probability = 0.60) exhibits the lowest average probability among the clusters, suggesting a relatively lower likelihood of antimicrobial activity for the peptides in this cluster. However, it is important to note that even though the average probability is lower, individual peptides within Cluster 4 might still possess significant antimicrobial capabilities.

Figure 3A depicts the proportional distribution of peptide secondary structures across various clusters. In Cluster 1, the β-strand secondary structure is predominant, accounting for 47%, followed by the α-helix structure at 21%. The presence of these alpha helices may suggest a tendency to form compact helical configurations, often linked to various biological functions [34,35,36]. In Cluster 2, the bridge structure is the most common, representing 33%. This protein structure is characterized by short amino acid segments that form hydrogen bonds with adjacent sections, offering stability, promoting specific molecular binding, and potentially indicating evolutionary conservation [37]. In this cluster, the β-strand secondary structure comes second, with the α-helix in third place. Conversely, Cluster 3 is primarily represented by the α-helix structure at 33%, followed by the β-strand, and in third place, the random coil structure. Unlike the ordered structures such as the α-helix and β-strand, the random coil represents protein segments that do not have a defined shape, potentially adopting multiple conformations. This structure can be influenced by various factors, including the presence of specific amino acids, molecular interactions, and environmental conditions like temperature, pH, pressure, and solute concentration [38,39]. Finally, Cluster 4 consists mainly of the α-helix structure, making up 70% in peptides primarily derived from amphibians such as brevinine, citropin, esculetin, magainin, phylloseptin, ranalexin, and temporin. These peptides exhibit an anticipated antimicrobial activity ranging from 0.96 to 1, and they show activity frequencies of proliferation (7 times), migration (8 times), angiogenesis (4 times), and immunomodulation (5 times).

Clusters 3 and 4 predominantly displayed α-helix secondary structures, which could indicate greater flexibility or capacity to interact with other peptides or molecules. The α-helix conformation, with its coiled and elongated structure, often allows for dynamic interactions due to its exposed side chains. This flexibility might be crucial for adaptive responses in various biological scenarios [40,41]. For instance, the external positioning of the side chains in the α-helix can foster interactions with lipid bilayers, making these peptides prime candidates for membrane penetration or association [42,43,44]. This trait is especially significant in the context of antimicrobial peptides, where disrupting or penetrating microbial membranes is a common mode of action [5].

The α-helix, a dominant motif in protein surface recognition, not only has unique properties that permit its existence in membranes but also enhances the peptide’s binding affinity with other proteins or ligands. This plays crucial roles in signaling pathways and enzymatic functions. Consequently, these characteristics have spurred the development of Helicon therapeutics, which can infiltrate cells and address interactions, like those between proteins, previously thought to be undruggable [36,45,46]. Their propensity to form coiled-coil motifs can also facilitate the dimerization or oligomerization of proteins, which can be essential for specific cellular functions or regulatory mechanisms. Considering the source of the peptides in Cluster 4, primarily from amphibians, it is worth noting that many amphibian-derived peptides are known for their antimicrobial properties [4,13]. The α-helical structure in these peptides might be an evolutionary adaptation to efficiently interact with and disrupt microbial membranes, providing the amphibians with a defense mechanism against potential pathogens [4]. The α-helix structures observed in Clusters 3 and 4 may have significant implications for wound healing. Peptides exhibiting this conformation can play pivotal roles in essential signaling pathways for wound repair, such as those promoting cell proliferation, angiogenesis, and collagen synthesis. Collagen, a fundamental constituent of the skin’s extracellular matrix, typically assumes a triple-helix structure [47,48]. The interaction of α-helical peptides with collagen may be vital in modulating its stability and overall arrangement during the wound-healing process [35]. Furthermore, these peptides can possess immunomodulatory attributes, crucial in regulating inflammation at wound sites [34,49]. A balanced inflammatory response is indispensable for optimal wound healing, preventing potential complications from excessive inflammation. Additionally, some of these peptides may interact with growth factors or their receptors, promoting wound recovery. The inherent scaffold of the α-helix conformation can augment these peptides’ binding capabilities and specificity [47,50,51].

Figure 3B illustrates the distribution of peptides based on their origin, grouped by clusters. From a closer look at this figure, the following insights emerge:

For Cluster 1:
-The predominant sequences in this cluster are from mammals, followed by those from plants and the amphibian category.-Among the peptides from mammals, we identify protegrin 1, cathelicidin 1, PDC213, and both beta and theta defensins, which primarily exhibit migration activities.-The plant-derived peptides include defensins from species such as *Allium sativum*, *Jatropha curcas*, and *Lupinus luteus*, which have wound-healing activities related to migration and immunomodulation.-Within the amphibian group, we notice peptides like odorranain B1, tigerinin, and taipehensin, which are specifically sourced from frogs of the Odorrana genus. It is a widely accepted fact that these frogs, especially in their dorsal skin, have glands producing a broad spectrum of peptides [12,13,52,53]. Applying these kinds of amphibian peptides directly to wounds has been shown to accelerate healing by promoting collagen production in fibroblasts, spurring keratinocyte proliferation, and increasing the levels of various growth factors beneficial for blood vessel formation [4,13].

Cluster 2: This cluster predominantly features peptides derived from mammals, closely followed by those from plants, reptiles, arthropods, and insects. The mammalian peptides in this cluster include alpha defensins, beta defensins, LL-37, lactoferricin, neutrophil defensins, and PR-39. The most common activities observed are immunomodulation and migration. Notably, human beta defensin 3 showcases all wound-healing activities. Within this cluster, plant defensins emerge as the dominant group, primarily exhibiting migration and immunomodulation activities, and can be traced back to species like *Brassica napus*, *Chassalia chartacea*, and *Viola abyssinica*. The reptilian peptides come from species such as *Bungarus fasciatus*, *Python bivittatus*, *Chelonia mydas*, and *Naja atra*, showcasing peptides like NA-CATH, SA-CATH, and cm-CATH. Most of these peptides have immunomodulatory activities, except for SA-CATH, which also promotes migration and angiogenesis. In the insect-derived peptide category, we identify species like *Bombyx mori, Hyalophora cecropia*, and *Copris tripartitus*. The peptides present include cecropins, hyphancin, and coprisin, which exhibit immunomodulatory, migration, and proliferation activities. It is noteworthy that cecropins are positively charged antimicrobial peptides that combat bacteria and fungi, exhibiting low toxicity towards eukaryotic cells. Their structure comprises N-terminal and C-terminal alpha helical sections connected by a flexible hinge region, which aids in their interaction with target microbes [54,55,56].

Cluster 3: This cluster predominantly features peptides from mammals, notably from humans and mice. This is followed by peptides sourced from plants and bacteria. The majority of human-derived peptides in this cluster include histatins, cathepsin, and theta defensin. Histatins mainly exhibit proliferation and migration activities [57]. Plant species represented in this cluster hail from *Cocos nucifera, Solanum tuberosum*, and *Tulipa gesneriana.*

Cluster 4: This cluster is characterized by a significant number of peptides from amphibians, with examples like brevinin, citropin, japonicin, magainin, esculetin, phylloseptin, and temporin. The antimicrobial peptide fragment esculentin-1a, from *Pelophylax lessonae*, aids wound healing, particularly in chronic skin ulcers. It stimulates cell migration, thus boosting wound epithelialization. Another set of peptides, temporins A and B, sourced from *Rana temporaria*, have been shown to decrease *Staphylococcus aureus* in cells, inducing cell proliferation which triggers wound closure. Cathelicidin-DM, from *Duttaphrynus melanostictus*, also facilitates wound healing, even more effectively than certain antibiotics like gentamicin [4]. Bacterial peptides, such as gramicidin A, cytolysin, pantocin, and subtilosin, come next, followed by those from fish, including epinecidin, myxinidin, and TP3. This cluster also has a higher concentration of synthetically derived peptides (H4, D2A21, K11) compared to the other clusters. The prominence of synthetic sequences in Clusters 3 and 4 indicates the significant influence of synthetic biology within the dataset, likely indicating a drive towards designing novel peptides with specific desired attributes [58,59,60].

### 2.2. Correlation Analysis between Physicochemical Properties and Peptide Activities

Figure 4 illustrates a correlation heatmap representing the relationship between various numerical features in the complete dataset. It was observed that peptide length exhibits a positive correlation with hydrophobicity and net charge at pH 7. This is visually depicted by the presence of red squares at the intersection of these characteristics. Additionally, hydrophobicity demonstrates a positive correlation with GRAVY and the percentage of hydrophobic residues. Antimicrobial prediction predominantly exhibits a positive correlation with peptide length and net charge at pH 7. On the other hand, certain features, such as the Boman index, appear to have weak correlations with other characteristics, as indicated by the light blue squares. In terms of wound-healing activities, proliferation displays a weak negative correlation with peptide length and hydrophobicity percentage. Similarly, migration exhibits a weak negative correlation with hydrophobicity. Conversely, angiogenesis does not exhibit any correlation with the physicochemical properties. However, immunomodulation demonstrates a positive correlation with peptide length.

### 2.3. Principal Component Analysis of Physicochemical Properties and Peptide Activity

Principal component analysis (PCA) is a dimensionality reduction technique that transforms our multidimensional dataset (in this case, peptide activities and physicochemical properties related to wound healing) into a set of new variables, called principal components, which are linear combinations of the original variables. The first principal component explains 36.99% of the variation in the data, while the second principal component explains 19.02% of the variation. Together, these two principal components account for approximately 56.01% of the total variation in the data. This allows for the visualization and analysis of the data structure in a two-dimensional space, as observed in Figure 5. The 44% of unexplained variability might be due to variables not captured in the analysis. These unmeasured properties could be toxicity, thermal stability, secondary structure, isoelectric point, adhesion properties, degradation rate, interaction with other compounds in the wound, amino acid sequence, solubility, and water-holding capacity, among others that are yet to be discovered [5,61,62,63].

The first principal component is strongly associated with proliferation, net charge, and peptide length. This suggests that peptides that are effective in promoting proliferation tend to exhibit notable influences from peptide length and net charge. This association indicates that proliferation, net charge, and peptide length are closely interconnected and play a fundamental role in the observed data variability. It implies that peptides promoting cellular proliferation have specific characteristics in terms of length and net charge.

Although the second principal component explains 19.02% of the variation, it also provides relevant information. In this case, immunomodulation is associated with peptide length, hydrophobicity, and net charge. The association between immunomodulation and peptide length suggests that the specific length of a peptide may impact its ability to modulate the immune response [64]. This implies that certain peptide sizes are more effective in modulating the immune system than others. Additionally, the association with hydrophobicity and net charge indicates that these physicochemical properties can play a significant role in the immunomodulatory activity of peptides [49,65]. Hydrophobicity can influence the peptide’s interaction with cell membranes and receptors, potentially affecting its capacity to modulate the immune response. On the other hand, net charge may be crucial for electrostatic interactions with cellular components, thus impacting the immune response [5].

### 2.4. Analysis of the Frequency and Distribution of Residues in Peptides

In the heatmap of Figure 6, it is observed that Cluster 1 has a high frequency of leucine (Leu) and isoleucine (Ile), which is a hydrophobic amino acid. This could suggest a predominance of hydrophobic interactions in this cluster’s peptides, potentially contributing to their membrane-penetrating abilities. However, Cys is a polar amino acid, similar to Arg in Cluster 1. These polar amino acids are often crucial for electrostatic interactions with membranes and receptors [66,67]. In Cluster 2, there is a high frequency of amino acids that include both polar residues such as Arg and Cys and nonpolar residues like Leu and Ala. Additionally, there is a notable presence of Gly, the smallest amino acid, which can provide flexibility to peptide chains and allow for specific conformations [68]. In Cluster 3, there is a notable abundance of polar amino acids such as Arg, Lys, and Cys, which are capable of participating in hydrogen bonding [67,69]. These amino acids contribute to enhanced solubility of peptides in aqueous environments. Lastly, Cluster 4, similar to Cluster 1, exhibits a predominant occurrence of apolar amino acids such as Leu and Ile. However, it is noteworthy that there is also a significant presence of polar amino acids like Lys and Phe. These polar amino acids can play a role in the stabilization of protein structures by participating in hydrogen bonding [33]. There is also a notable presence of Gly in Cluster 4.

In summary, the heatmap indicates a prevalent presence of hydrophobic amino acids, mainly Leu, Gly, and Ile, in antimicrobial peptides across all clusters. Additionally, there are observable occurrences of polar amino acids such as Arg, Lys, Phe, and Cys. This might be related to the known ability of these peptides to interact with and disrupt lipid membranes, a common mechanism of antimicrobial action. At the same time, the presence of polar and small amino acids in some clusters could reflect the need for these peptides to be soluble in aqueous environments and to adopt specific conformations for interacting with their targets.

We also analyzed the presence of the most frequent amino acids in the N-terminal region, the central region, and the C-terminal region in order to elucidate their relationship with antimicrobial and wound-healing activity (see Table 1). For Cluster 1, the peptides in this cluster have alanine, leucine, and glycine in the N-terminal region. This combination of hydrophobic amino acids and a small, flexible amino acid could facilitate the initial interaction with lipid bilayers, an important step in antimicrobial activity. At the central region level in Cluster 1, glycine, alanine, and leucine were found. This combination is similar to that of the N-terminal region, which could imply a similar structure in these two regions. In the C-terminal region, the peptides have lysine, alanine, and arginine. Positively charged amino acids like lysine and arginine can facilitate interaction with negatively charged bacterial membranes, possibly increasing antimicrobial activity [5,66]. For Cluster 2, in the N-terminal region, peptides contain glycine, alanine, and proline. The presence of proline, an amino acid that can induce turns and bends in the peptide structure, could imply a unique structural conformation in this region [68]. In the central region, peptides contain leucine, glycine, and serine. The presence of serine, a polar amino acid, could facilitate interactions with the aqueous environment. For the C-terminal region of Cluster 2, the peptides contain leucine, proline, and serine. This combination of hydrophobic, polar, and turn-inducing amino acids could result in a unique structural diversity. In the N-terminal region of Cluster 3, peptides contain glycine, serine, and alanine. This blend of polar and nonpolar amino acids could suggest a variety of potential interactions with lipid bilayers and the aqueous environment [69]. The central region carries a similar composition to the N-terminal region, with peptides containing glycine, alanine, and serine. This correspondence hints at possible symmetry in the structural and functional properties across these regions. As for the C-terminal region, peptides contain serine, alanine, and glycine, with serine—a polar amino acid—being predominant. This prevalence could facilitate interaction with the aqueous environment and contribute to the solubility of the peptides, implying a specific adaptability for different biological environments and conditions [70]. Finally, in Cluster 4 the N-terminal region peptides comprise glycine, alanine, and leucine. This particular combination could potentially enhance the interaction with lipid bilayers and impart stability to the peptide structure [5]. The central region peptides are composed of leucine, glycine, and serine. Here, the presence of serine could facilitate the solubility of the peptides and their interaction with their environment [61,71]. Finally, in the C-terminal region, the peptides include glycine, serine, and alanine. The presence of glycine and serine might augment the flexibility and solubility of the peptides, while alanine could contribute to structural stability [68]. These combined properties could play crucial roles in the peptides’ function and efficacy in their respective biological roles.

Overall, a combination of hydrophobic amino acids, amino acids capable of adopting specific conformations, and amino acids with charged properties in the N-terminal, central, and C-terminal regions of the peptides in each cluster is observed. These characteristics can contribute to the structure, function, and physicochemical properties of the peptides in each cluster. In a biological context, these results could be especially relevant in the field of antimicrobial peptide research and peptide engineering for wound healing. The amino acid patterns observed in the different clusters could be contributing to the antimicrobial activity of these peptides in various ways. For example, the presence of charged amino acids in the terminal regions may enhance the peptide’s interaction with negatively charged bacterial membranes [42,72]. Moreover, some peptides have also shown promise for wound healing. In this context, the composition and sequence of amino acids can influence the peptide’s ability to interact with cells and proteins in the wound, promoting healing. Additionally, the different clusters could represent different families of peptides with similar structures and functions. This could be useful for predicting the properties of new peptides based on their similarity to the peptides in these clusters.

### 2.5. Analysis of the Motifs and Family Domains in Peptides

Structural motifs in peptides refer to recurring patterns or configurations in their sequence of amino acids that contribute to their three-dimensional conformation and biological function [73]. These structural motifs are crucial in determining how peptides fold and interact with other molecules in their cellular environment. When analyzing the most frequent motifs found in each cluster, the following was discovered:Cluster 1: The most frequent motifs are ‘CRC’, ‘RCI’, ‘CIC’, ‘CTR’, and ‘GFC’.Cluster 2: The most frequent motifs are ‘GTC’, ‘CCR’, ‘YCR’, ‘CRR’, and ‘CYC’.Cluster 3: The most frequent motifs are ‘SHR’, ‘CRC’, ‘KFH’, ‘FHE’, and ‘HEK’.Cluster 4: The most frequent motifs are ‘KKF’, ‘GGL’, ‘KKL’, ‘SLI’, and ‘FKK’.

These motifs could potentially play significant roles in the biological activity of the peptides in each cluster, as they are recurrent patterns in the amino acid sequences. Motifs can influence the structure, stability, and function of peptides, including their interactions with other molecules [73,74]. For instance, motifs rich in cysteine (C) residues, such as ‘CRC’ in Clusters 1 and 3, could potentially form disulfide bridges that contribute to the stability of the peptide structure [69]. Similarly, motifs rich in basic residues like lysine (K), such as ‘KKF’ and ‘KKL’ in Cluster 4, could facilitate interactions with negatively charged microbial membranes, a common mechanism of antimicrobial peptides [66].

Regarding the functional domains identified, according to Table 2, there is a high frequency of the cecropin peptide family, followed by defense response domains belonging to beta defensins.

Upon conducting a filtration of the peptide database, our primary focus was centered on peptides exhibiting an antimicrobial activity of 1.0 based on deep learning antimicrobial prediction, in addition to encompassing all wound-healing activities. The results led to the identification of two peptides, namely human beta defensin 3 (HBD-3) and nisin A (see Table 3), both of which are categorized within Cluster 2. HBD-3 is characterized by a sequence length of 45 residues and a hydrophobicity value of 32.69. The peptide originates from human skin, tonsils, and saliva [75]. HBD-3 stimulates human fibroblasts, promoting cell migration, proliferation, and the production of angiogenic growth factors via the FGFR1/JAK2/STAT3 signaling pathway. In vivo, HBD-3 accelerates wound healing, increasing neutrophils, macrophages, fibroblasts, and angiogenesis, making it a potential candidate for wound treatment [75]. In a similar vein, nisin A, with a sequence length of 34 residues and a hydrophobicity value of 32.38, also belongs to Cluster 2 [76]. This peptide is derived from *Lactococcus lactis* [77].

### 2.6. Final Findings

Antimicrobial and wound-healing peptides have garnered increasing interest in the scientific community due to their potential as therapeutic agents in treating infections and chronic injuries, such as those caused by burns and diabetes [1]. As their names suggest, antimicrobial peptides can kill or inhibit the growth of microorganisms, while wound-healing peptides can promote wound healing by modulating various cellular responses, including migration, proliferation, immunomodulation, and angiogenesis [2,49]. Designing a peptide with both characteristics represents a significant advantage. This complex functionality is intrinsically related to various physicochemical characteristics of peptides, such as their length, net charge, and hydrophobicity, among others. These properties make them highly promising candidates for the development of novel treatments [10,11,29].

The diversity of antimicrobial and wound-healing peptides obtained from our database, distributed into four clusters with unique physicochemical and biological properties, reflects the structural and functional richness of these peptides. Specifically, our results highlight the importance of peptide length, hydrophobicity, and net charge in determining their antimicrobial and wound-healing capabilities [5,8]. For instance, peptides in Cluster 2 are on average the longest and have the highest net charge, potentially facilitating their interaction with polar environments. Moreover, these peptides are primarily associated with proliferation activity, a crucial step in wound healing [78,79]. In contrast, peptides in Cluster 3, which are shorter and less hydrophobic, exhibit a higher proportion of migration activity, essential for wound closure and tissue regeneration. Furthermore, our correlation analysis reveals that peptide length has a positive correlation with hydrophobicity and net charge, which is important for antimicrobial activity [5].

PCA also underscores the importance of peptide length and net charge, strongly associated with proliferation and immunomodulation activities. This suggests that the physicochemical characteristics of peptides can play a significant role in modulating the immune response and promoting cell proliferation [34,49,64,76].

However, this study has some limitations: (1) The dataset used for analysis may not fully represent the diversity of all peptides with dual wound-healing and antimicrobial activities. More extensive and diverse databases would enhance the reliability of the conclusions. (2) Artificial intelligence algorithms used to predict antimicrobial activity may have limitations in accurately predicting peptide biological activity. (3) The study focused on in silico analysis and did not include experimental validation of the identified peptide activities. This study serves as a preliminary exploration to understand the characteristics of peptides with dual antimicrobial and wound-healing activity. It is essential to consider these limitations when interpreting the results and designing future research in this field.

Furthermore, this study represents a crucial first step towards the development of a research project aimed at bioprospecting and designing peptides with dual activity in transcriptomes of species such as amphibians, reptiles, and insects, using the sequences collected in this investigation [12,53].

Recent investigations have elucidated the multifaceted attributes of peptides, highlighting their dual activity and unique structural and functional properties [8]. This understanding serves as a foundational platform for advancing bioprospecting initiatives. As the field gains insights into the physicochemical determinants of peptide-mediated biological activities, there is potential to engineer peptides with tailored characteristics for specific therapeutic applications. Leveraging advancements in artificial intelligence and machine learning, predictive modeling of peptide activity based on sequence and structural data is becoming increasingly sophisticated, though empirical validation remains essential for translational purposes [29,31,60,73].

The diverse origins of peptides, ranging from amphibians to insects and mammals, suggest a substantial, untapped reservoir for bioprospecting [13,56,65,80]. By capitalizing on the intrinsic properties of these peptides, researchers can target natural sources poised to yield peptides with analogous or superior traits. An integrative approach, encompassing both naturally derived and synthetic peptides, is postulated to expedite the discovery of novel dual-activity compounds [5,7,8]. While naturally derived peptides, products of evolutionary processes, are renowned for their specificity and potentially reduced side effects, they present challenges [59]. These include the intricacies of extraction and purification, as well as inherent variability in peptide properties based on their source [12]. Moreover, sustainable bioprospecting practices are imperative to mitigate potential ecological repercussions, especially concerning vulnerable species [81,82].

In the realm of wound healing, the integration of peptides with materials such as hollow silica nanoparticles has demonstrated enhanced healing outcomes. From an analytical perspective, peptides are instrumental in decoding wound repair mechanisms, with a particular focus on signaling pathways like MAPK and TGF-β1/SMAD. Additionally, researchers are leveraging peptides as molecular probes to delve into the emerging competitive endogenous RNA (ceRNA) mechanism, shedding light on RNA interactions [4]. This research has the potential to uncover the roles of miRNAs in wound recovery, underscoring the dual therapeutic and analytical potential of peptides. Ultimately, the therapeutic potential of antimicrobial and wound-healing peptides is just beginning to be discovered, and their study will continue to unlock new pathways for the treatment of infections and wounds.

## 3. Materials and Methods

### 3.1. Selection of Peptide Databases

The search was initially performed in the databases listed in Table 4. Following that, a comprehensive search was conducted in recent scientific articles to identify peptides with properties related to wound-healing capacity, angiogenesis, migration, proliferation, and immunomodulation. As a result, a database was compiled consisting of 207 peptides with antimicrobial and wound-healing activities. These activities were verified through laboratory tests carried out by the authors of the peptides themselves.

### 3.2. Search and Estimation of Physicochemical Properties

The database was stored in a spreadsheet (see Appendix A), which included values for variables such as net charge at pH 7, hydrophobicity, sequence length, GRAVY value, Boman index, solubility, origin species of each peptide, and secondary structure prediction for each peptide. For peptides where not all the required information was available in the databases mentioned in Table 4, estimates were made using other tools indicated in Table 5. The secondary structure was predicted using NP: network protein sequence analysis (https://npsa-prabi.ibcp.fr/cgi-bin/npsa_automat.pl?page=/NPSA/npsa_server.html, accessed on 12 April 2023) with the MLRC method [87]. For the prediction of antimicrobial activity, AMPlify (https://github.com/bcgsc/AMPlify, accessed on 12 April 2023) was used, which is a deep learning-based tool [88].

### 3.3. Statistics and Classification of Wound-Healing Peptides

Descriptive statistical analysis is conducted on the 207 peptides, along with clustering analysis, correlation analysis, and PCA. The classification analysis (K-means) helps identify groups of peptides with similar characteristics, while correlation analysis and PCA provide insights into the relationship and relative importance of physicochemical properties concerning wound-healing activities [9,93]. The K-means artificial intelligence algorithm is employed to classify the wound-healing peptides based on their physicochemical characteristics, such as net charge at pH 7, hydrophobicity percentage, sequence length, and GRAVY value. This algorithm identifies clusters within the dataset by grouping peptides with similar properties. The resulting clustering information is then incorporated into the peptide database, assigning each peptide to its respective cluster [9,94]. A one-way ANOVA is carried out using SPSS version 27 (https://www.ibm.com/support/pages/downloading-ibm-spss-statistics-27, accessed on 20 April 2023) [95]. Subsequently, a Tukey post hoc test is applied to identify differences among the treatment groups. The hypotheses are established based on the population means of the dependent variable(s) across distinct groups, namely clusters (see Appendix B). Results with a *p*-value below 0.05 are considered statistically significant [96].

In addition to the clustering analysis, correlation analysis is utilized to examine the association between physicochemical properties and wound-healing activities of the peptides. A common method of correlation analysis is the Pearson correlation coefficient, which quantifies the linear relationship between two variables [97]. The correlation coefficient is calculated between each pair of physicochemical properties and wound-healing activities. A correlation coefficient close to 1 indicates a strong positive correlation, while a coefficient close to −1 indicates a strong negative correlation. A coefficient close to 0 indicates a weak or non-existent correlation.

Another technique employed is PCA, which is a dimensionality reduction method. PCA allows reducing a set of correlated variables to a smaller set of uncorrelated principal components. These principal components capture most of the variability in the original data. By applying PCA to the physicochemical properties of the peptides, one can determine which variables contribute most to the total variability and how they relate to each other [98]. This aids in identifying the most relevant properties in terms of their association with the wound-healing activities of the peptides.

### 3.4. Functional Domain Analysis

To analyze the functional domains of the 207 peptides with antimicrobial and wound-healing activity, InterPro (https://www.ebi.ac.uk/interpro/search/sequence/, accessed on 20 April 2023) was used. InterPro is a database that integrates information from various sources on protein domains, families, and binding sites. The analysis of functional domains provided a deeper understanding of the characteristics and potential mechanisms of action of the studied peptides, contributing to the elucidation of their antimicrobial and wound-healing activity [12].

## 4. Conclusions

This study provides valuable insights into the relationship between the physicochemical properties of peptides, wound healing, and the antimicrobial activities of peptides with dual functionality. Moreover, it allowed for the identification of patterns such as motifs or functional domains and their distribution throughout the peptide sequence. Similarly, relevant species for the bioprospecting of future peptides were identified. The findings of this work can guide future research and peptide engineering efforts aimed at developing new therapeutic agents for wound healing and combating microbial infections. However, further experimental validations and clinical research are needed to translate these findings into practical applications in the biomedical field.

## Figures and Tables

**Figure 1 ijms-24-13091-f001:**
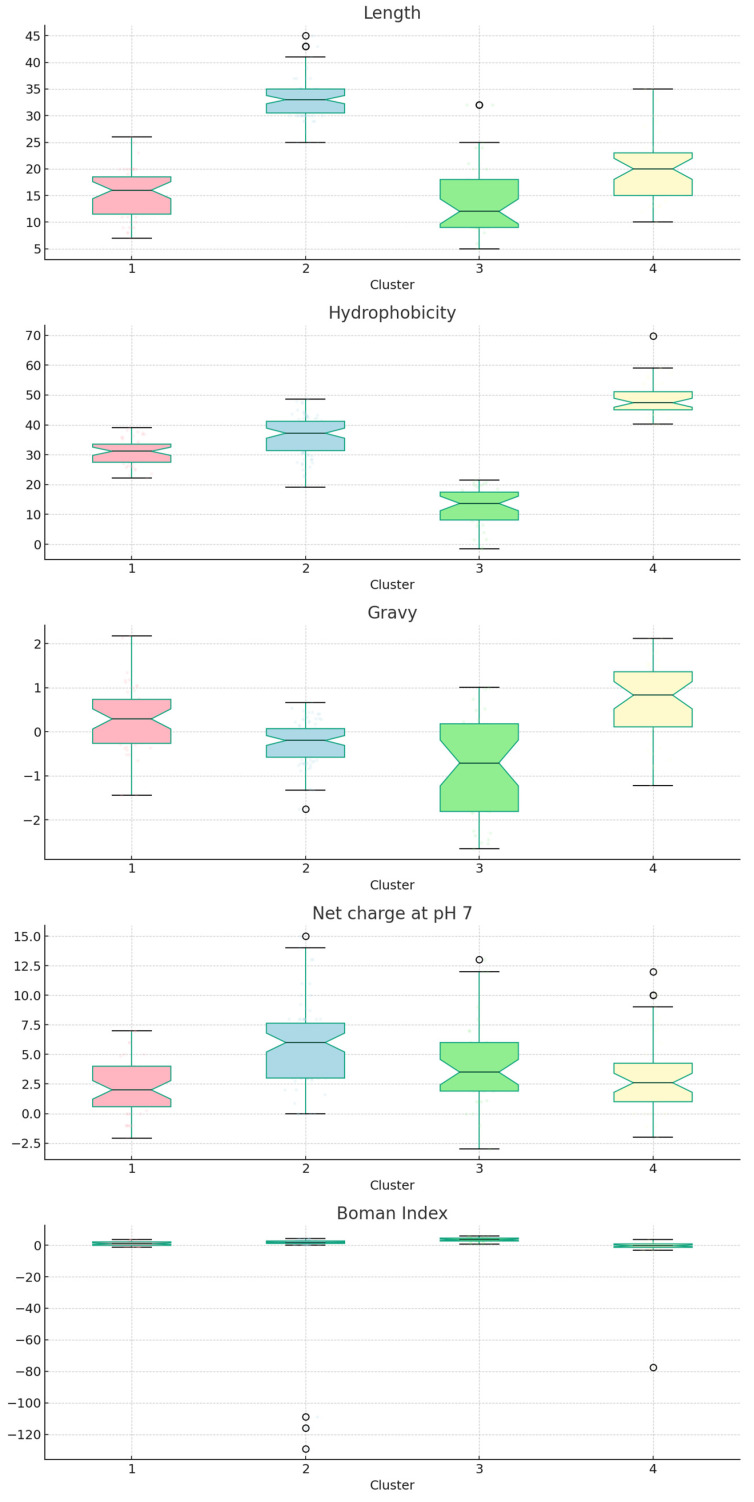
Distribution of peptide properties among four clusters.

**Figure 2 ijms-24-13091-f002:**
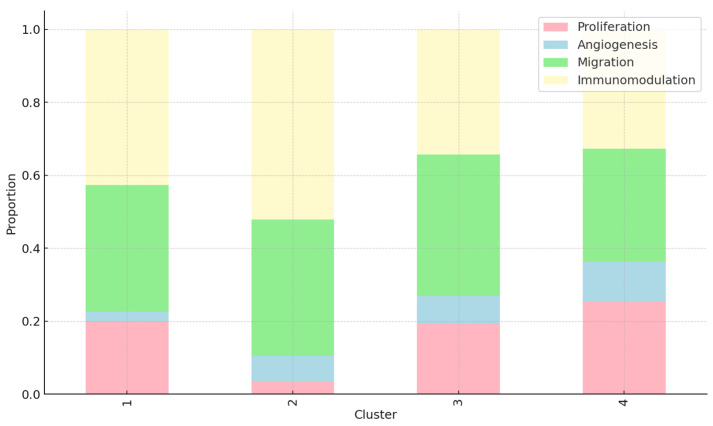
Comparison of wound-healing activities by cluster.

**Figure 3 ijms-24-13091-f003:**
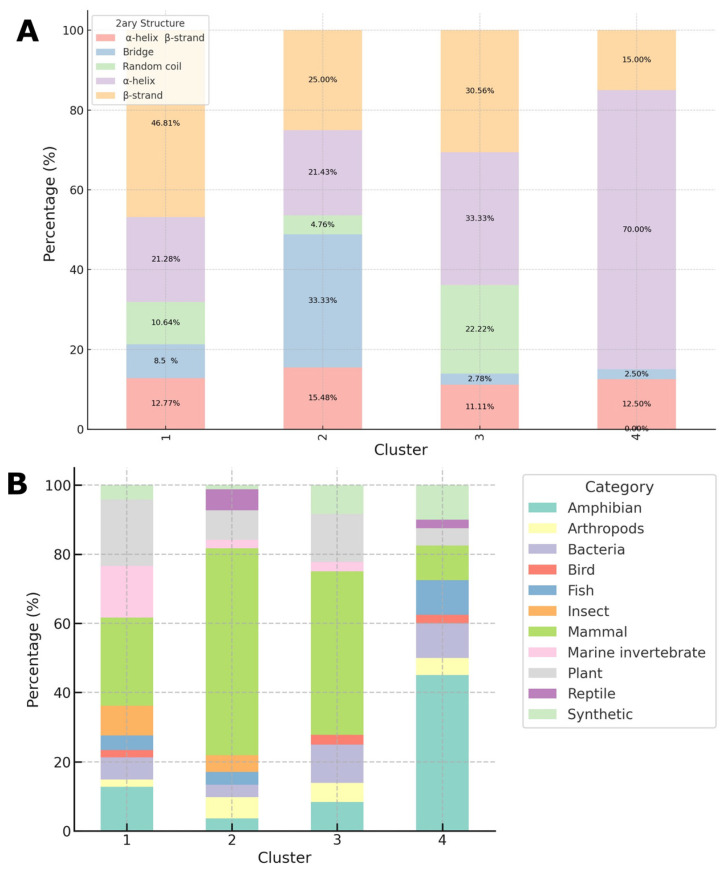
Comparison profile of sources of peptides and secondary structures in different clusters. (**A**) Profile secondary structures by cluster. (**B**) Profile of sources from animal categories by cluster.

**Figure 4 ijms-24-13091-f004:**
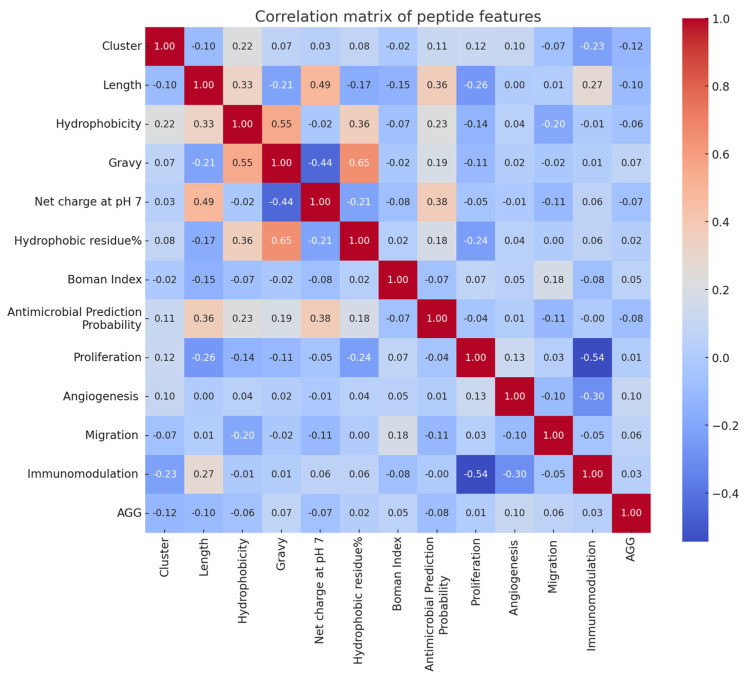
Correlation matrix between peptide activities and physicochemical properties. The cells in the heatmap are colored according to the correlation value between the corresponding features: Positive values are in warmer tones (red) and negative values are in cooler tones (blue).

**Figure 5 ijms-24-13091-f005:**
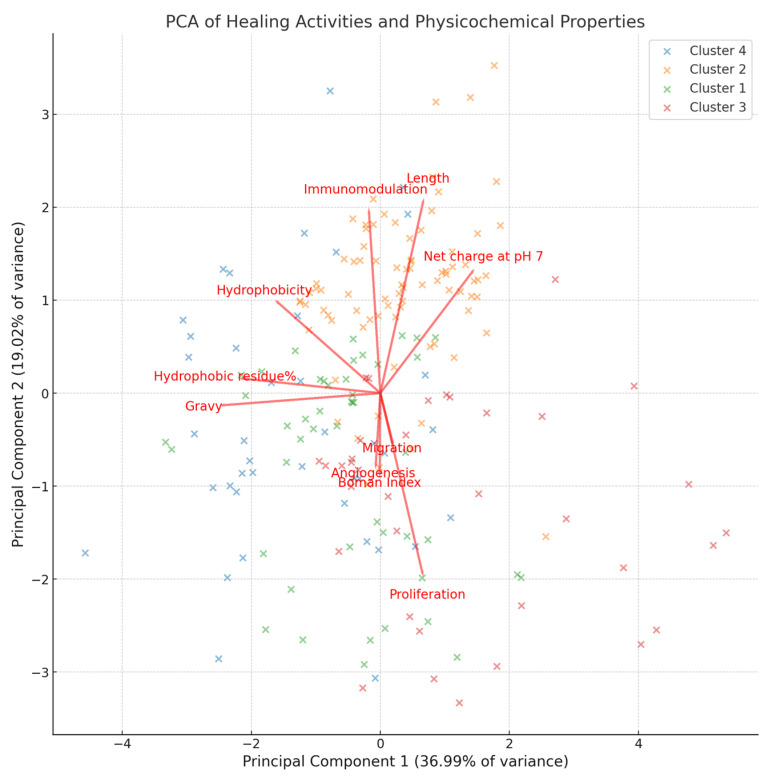
PCA of wound-healing activities and physicochemical properties of antimicrobial peptides.

**Figure 6 ijms-24-13091-f006:**
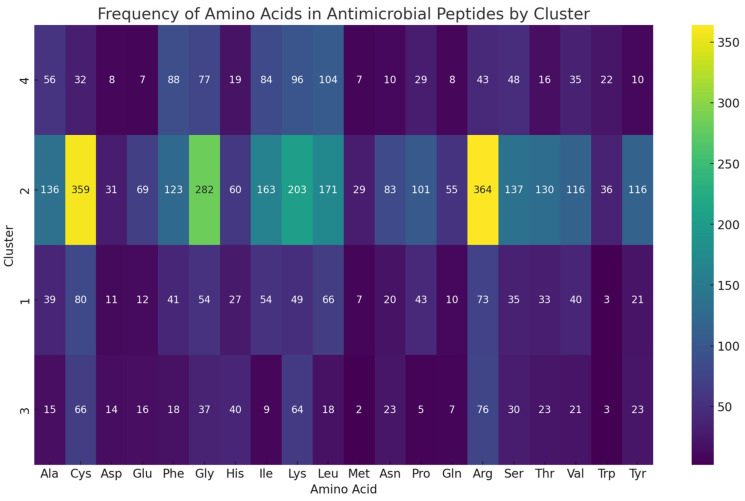
Heatmap of most frequents residues by cluster.

**Table 1 ijms-24-13091-t001:** Residue frequency according to peptide region.

Cluster	N-Terminal Region	Central Region	C-Terminal Region
Cluster 1	Ala, Leu, Gly	Ala, Leu, Gly	Lys, Ala, Arg
Cluster 2	Gly, Ala, Pro	Leu, Gly, Ser	Leu, Pro, Ser
Cluster 3	Gly, Ser, Ala	Gly, Ser, Ala	Gly, Ser, Ala
Cluster 4	Gly, Ala, Leu	Leu, Gly, Ser	Gly, Ser, Ala

**Table 2 ijms-24-13091-t002:** Functional domains most frequently found in peptides.

GO Terms	Family Domain	Frequency
Antibacterial humoral response (GO:0019731)	IPR000875 Cecropin 3–32	6
Extracellular region (GO:0005576)		
Defense response (GO:0006952)	IPR001855	2
	Defensin_beta-typ	
	ene-31	
Defense response (GO:0006952)	IPR001855	2
	Defensin_beta-typ	
	mar-30	
Defense response (GO:0006952)	IPR001855	2
	Defensin_beta-typ	
	abr-39	
Defense response (GO:0006952)	IPR017982: Defensin_insect	2
	nov-20	
	IPR017982: Defensin_insect	
	29–38	

**Table 3 ijms-24-13091-t003:** Comparison of properties between nisin A and human beta defensin 3 (HBD-3).

Name	Sequences	Cluster	Lenta	Hydrophobicity	GRAVY	Net Charge	2ary Structure	Species
Nisin A	ITSISLCTPGCKTGALMGCNMKTATCHCSIHVSK	2	34	32.38	0.41	3	Random coil	*Lactococcus lactis*
HBD-3	GIINTLQKYYCRVRGGRCAVLSCLPKEEQIGKCSTRGRKCCRRKK	2	45	32.69	−0.70	11	α-helix, β-strand	*Homo sapiens*

**Table 4 ijms-24-13091-t004:** Databases of antimicrobial peptides.

Database Names	URL/Citation
APD3	https://aps.unmc.edu/ (accessed on 12 April 2023) [83]
DBAASP	https://dbaasp.org/home (accessed on 12 April 2023)
CAMP_R4_	http://www.camp.bicnirrh.res.in/index.php/ (accessed on 12 April 2023) [84]
YADAMP	http://yadamp.unisa.it/default.aspx/ (accessed on 12 April 2023) [85]
DRAMP	http://dramp.cpu-bioinfor.org/ (accessed on 12 April 2023) [86]

**Table 5 ijms-24-13091-t005:** Sources used for the estimation of physicochemical properties.

Tool Name	Description	URL	Citation
PepCalc	Peptide properties calculator	https://pepcalc.com/ (accessed on 12 April 2023)	[89]
Peptide Analyzing Tool Thermo Scientific	Peptide synthesis and proteotypic peptide analysis tool	https://www.thermofisher.com/co/en/home.html (accessed on 12 April 2023)	[90]
Tango	A computational algorithm for the prediction of aggregated regions in unfolded polypeptide chain	http://tango.crg.es/about.jsp (accessed on 12 April 2023)	[32]
BACHEM	Peptide properties calculator	https://www.bachem.com/knowledge-center/peptide-calculator/ (accessed on 12 April 2023)	[91]
Quick2D	Bioinformatics toolkit	https://toolkit.tuebingen.mpg.de/tools/quick- (accessed on 12 April 2023)	[92]

## Data Availability

Data available in manuscript and the Appendix A.

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
