# Peer review of "Characterization and Classification In Silico of Peptides with Dual Activity (Antimicrobial and Wound Healing)"

_ijms, 2023, doi:10.3390/ijms241713091_

Round 1
Reviewer 1 Report
In this in silico investigation, the authors utilized bioinformatic tools and statistical techniques to explore dual-function peptides with potential applications in addressing chronic wounds and antibiotic resistance. Overall the study provides a foundation for therapeutic development in this field. Therefore, with the following minor revisions, this paper can be accepted:
1. While the first and second principal components explain a substantial portion of the variation (56.01%), there are likely additional principal components contributing to the remaining variation. Could authors discuss the reasons for focusing solely on the first two components in the analysis?
2. The data presented in figure 3B doesn’t align with the description in lines 169-182. Specifically, it seems that Cluster 1 exhibits more b-stand characteristics than a-helix, while Cluster 4 displays a higher proportion of α-helix. To enhance clarity and distinguish the findings better, the authors should consider providing the percentage values for each color bar in the figure.
3. The association between peptide characteristics (length, net charge, etc.) and cellular proliferation is an interesting finding. Could the authors discuss a little more about the potential biological mechanisms or existing literature supporting the relationship between these physicochemical properties and cellular proliferation?
4. Considering the vast reservoir of potential peptides from various sources, including amphibians, insects, and mammals, could the authors discuss how their research findings could inspire and guide future bioprospecting efforts to discover novel peptides with dual activity? It would be beneficial to discuss the advantages and challenges associated with naturally derived peptides compared to synthetic peptides.
Author Response
I am writing to express my heartfelt gratitude for your thorough review and valuable suggestions for my manuscript entitled "Characterization and Classification in silico of Peptides with Dual activity (Antimicrobial and Wound Healing)" submitted to the International Journal of Molecular Sciences.
Your meticulous feedback provided insightful perspectives that have significantly enhanced the quality and clarity of the manuscript. Your constructive criticism not only shed light on the areas that required further elucidation but also helped strengthen the work's scientific merit.
1.Although the first two principal components account for a substantial portion of the variation (56.01%), there are likely additional principal components that contribute to the remaining variation. Could the authors discuss the reasons for focusing solely on the first two components in the analysis?
Response:
The first principal component, which accounts for 36.99% of the variation, is strongly associated with proliferation, net charge, and peptide length. This relationship suggests that peptides effective in promoting proliferation tend to exhibit specific characteristics in terms of length and net charge. In other words, there's an intrinsic interrelation between cell proliferation and these physicochemical properties of peptides.
Although the second principal component explains a smaller percentage of the variation (19.02%), it is no less significant. This component highlights the relationship between immunomodulation and properties such as peptide length, hydrophobicity, and net charge. The association between immunomodulatory activity and peptide length suggests that certain peptide lengths might be inherently more effective in modulating the immune system. Moreover, hydrophobicity and net charge are physicochemical properties that can influence how a peptide interacts with cells and, therefore, its ability to modulate the immune response.
Given that these two principal components capture more than half of the total variability of the dataset, their analysis provides substantial insight into the properties and activities of peptides in relation to wound healing. The 44% of unexplained variability might be due to variables not captured in the analysis. These unmeasured properties could be toxicity, thermal stability, secondary structure, isoelectric point, adhesion properties, degradation rate, interaction with other compounds in the wound, amino acid sequence, solubility, water holding capacity, among others that are yet to be discovered [5,45–47].
- The data presented in Figure 3B do not align with the description on lines 169-182. Specifically, it appears that Cluster 1 exhibits more characteristics of a beta-helix structure than of an alpha-helix, whereas Cluster 4 shows a higher proportion of alpha-helix. To enhance clarity and better differentiate the findings, the authors should consider providing the percentage values for each color bar in the figure.
Response: The suggested modifications were made in the text and highlighted in yellow.
3.The association between peptide features (length, net charge, etc.) and cell proliferation is an intriguing finding. Could the authors elaborate further on the potential biological mechanisms or existing literature that supports the relationship between these physicochemical properties and cell proliferation?
Response: this was added to the manuscript
The ANOVA analysis revealed significant differences in various physicochemical parameters between the clusters. Specifically, length and hydrophobicity stand out as key distinguishing factors among the groups. These differences might be linked to the peptides' ability to interact with cellular membranes and with receptors involved in healing processes such as Epidermal Growth Factor Receptor (EGFR), Vascular Endothelial Growth Factor Receptor (VEGFR), and Transforming Growth Factor Beta (TGFB), which are essential for their antimicrobial and healing activity. The EGFR signaling pathway plays a crucial role in promoting epidermal cell growth. This pathway's ability to stimulate HaCaT cell migration is potentially linked to the transactivation of the epidermal growth factor receptor (EGFR) and the signaling intermediates ERK1/2 and Smad2. Other studies have pointed to the PI3K/AKT and JNK pathways as vital for cell migration. In the context of HSF cells, the NF-κB and ERK pathways likely influence migration capabilities [14,19].
TGF-β is a pivotal growth factor essential for wound healing, especially its subtypes TGF-β1, 2, and 3. During trauma's acute phase, TGF-β1 is released, aiding in the chemotaxis of macrophages and fibroblasts to the wound and bolstering keratinocyte proliferation. TGF-β3 is also influential in regulating cell migration. The downstream effector of TGF-β, SMAD, plays a role in the TGF-β/SMAD signaling pathway, promoting skin angiogenesis, wound contraction, and reducing inflammation [14,20,21]. Both MAPK and NF-κB signaling pathways intersect in wound repair and are closely tied to inflammation. Modulating their phosphorylation can mitigate inflammatory responses during wound healing. There's also evidence of an interplay between the TGF-β and MAPK pathways [14,22,23]. Macrophages are integral throughout the wound healing process, from inflammation to tissue reconstruction. Recent studies have emphasized the importance of TNF and TGF-β1, predominantly produced by macrophages, in the wound healing process. In the inflammation phase's early stages, TNF is up-regulated, recruiting inflammatory cells like macrophages. These cells then release more TGF-β1 to the wound sites, facilitating wound healing due to its chemotactic properties. TGF-β1 also orchestrates cellular proliferation, migration, and granulation tissue regeneration [14,22,23].
Literature related to peptides and proliferation:
https://www.frontiersin.org/articles/10.3389/fphar.2023.1120228/full
https://www.ncbi.nlm.nih.gov/pmc/articles/PMC6430730/
https://www.ncbi.nlm.nih.gov/pmc/articles/PMC4789108/
- Considering the vast reservoir of potential peptides from diverse sources, including amphibians, insects, and mammals, could the authors discuss how the findings from their research might inspire and guide future bioprospecting efforts to discover new peptides with dual activity? It would be beneficial to discuss the advantages and challenges associated with naturally derived peptides compared to synthetic peptides.
Response:
Our research has identified specific structural and functional characteristics in the peptides we studied, which display dual activity. These findings can serve as a foundation or reference for future bioprospecting efforts. By better understanding the properties and mechanisms of action of these peptides, we can direct bioprospecting efforts towards natural sources that are more likely to contain peptides with similar characteristics. We believe that a combination of approaches, using both natural and synthetic peptides, could be the key to discovering new compounds with dual activity.
Naturally derived peptides offer certain advantages and challenges compared to synthetic peptides:
Advantages:
- Evolutionary Diversity: Natural peptides have evolved over millions of years, which may have optimized their function and effectiveness in biological interactions.
- Specificity: Natural peptides often show high specificity for their targets due to coevolution with their ligands or receptors.
- Lower Toxicity: Being natural products, some peptides may exhibit fewer side effects compared to synthetic compounds.
Challenges:
- Isolation and Purification: Extracting and purifying peptides from natural sources can be laborious and expensive.
- Variability: There might be variations in the peptide's structure and function depending on the source and growth conditions.
- Conservation: Bioprospecting can threaten rare or endangered species if not carried out sustainably.

Reviewer 2 Report
Dear Authors, i believe that your ms is very good but us reader i cannot understand the section 2. You must seperate results and discussion. Secondly you must reference in the Introduction. Furthermore, I recommend that the authors reorganize the discussion section. The obtained results in this research are valuable and interesting, but they are challenging to understand in the current form of the discussion. My suggestion is to first explain the most significant findings of this experiment and then compare them with existing research.
The statistical approach is poor. You have a lot of data to do a proper analyses.
Thank you for the opportunity to review this paper. I have thoroughly read the manuscript and provided major comments that need to be addressed in the pdf file.

The English language of the manuscript needs to be revised in places.
Author Response
I am writing to express my heartfelt gratitude for your thorough review and valuable suggestions for my manuscript entitled "Characterization and Classification in silico of Peptides with Dual activity (Antimicrobial and Wound Healing)" submitted to the International Journal of Molecular Sciences.
Your meticulous feedback provided insightful perspectives that have significantly enhanced the quality and clarity of the manuscript. Your constructive criticism not only shed light on the areas that required further elucidation but also helped strengthen the work's scientific merit.
Dear authors, I believe your manuscript is very good, but as a reader, I cannot understand Section 2. You should separate the results and discussion. Secondly, references are needed in the Introduction. Furthermore, I recommend that the authors reorganize the discussion section. The results obtained in this research are valuable and intriguing, but they are difficult to comprehend in the current format of the discussion. My suggestion is that you first explain the most significant findings from this experiment and then compare them to existing research.
Response: We appreciate the suggestion to divide the manuscript into separate results and discussion sections. However, given the interconnected structure of our findings and the thematic coherence we aimed to maintain, we believe that splitting them into two distinct sections might disrupt the narrative flow and potentially diminish the clarity and impact of our conclusions. We've endeavored to present a cohesive narrative that integrates both our findings and their interpretation in an understandable format. Nonetheless, we value your feedback and will consider how to enhance the clarity of the section in question to ensure readers can easily follow our insights.
The statistical approach is lacking. You have ample data for a proper analysis.
Response: One way ANOVA was performed in the manuscript, in order to improve statistical analysis between each cluster result.
Thank you for the opportunity to review this article. I have read the manuscript thoroughly and provided significant comments that should be addressed in the PDF file.
Response: In accordance with the suggested comments, the manuscript has been corrected and improved.
